# Personalized bacteriophage therapy to treat pandrug-resistant spinal *Pseudomonas aeruginosa* infection

T. Ferry[1,2,3,4 ✉], C. Kolenda[2,3,4,5], F. Laurent[2,3,4,5], G. Leboucher [6], M. Merabischvilli[7], S. Djebara[7], C.-A. Gustave[2,4,5], T. Perpoint[1], C. Barrey[8], J.-P. Pirnay [7,10] & G. Resch[9,10]

Bone and joint infections (BJI) are one of the most difficult-to-treat bacterial infection, especially in the era of antimicrobial resistance. Lytic bacteriophages (phages for short) are natural viruses that can selectively target and kill bacteria. They are considered to have a high therapeutic potential for the treatment of severe bacterial infections and especially BJI, as they also target biofilms. Here we report on the management of a patient with a pandrug-resistant *Pseudomonas aeruginosa* spinal abscess who was treated with surgery and a personalized combination of phage therapy that was added to antibiotics. As the infecting *P. aeruginosa* strain was resistant to the phages developed by private companies that were contacted, we set up a unique European academic collaboration to find, produce and administer a personalized phage cocktail to the patient in due time. After two surgeries, despite bacterial persistence with expression of small colony variants, the patient healed with local and intravenous injections of purified phages as adjuvant therapy.

[1] Service de Maladies Infectieuses et Tropicales, Hôpital de la Croix-rousse, Hospices Civils de Lyon, Lyon, France. [2] Université Claude Bernard Lyon 1, Villeurbanne, France. [3] Centre de Référence des Infections Ostéo-Articulaires Complexes de Lyon (CRIOAc Lyon), Hospices Civils de Lyon, Lyon, France. [4] Centre International de Recherche en Infectiologie, CIRI, Inserm U1111, CNRS UMR5308, ENS de Lyon, UCBL1 Lyon, France. [5] Institut des Agents Infectieux, Hôpital de la Croix-rousse, Hospices Civils de Lyon, Lyon, France. [6] Pharmacie Hospitalière, Hôpital de la Croix-rousse, Hospices Civils de Lyon, Lyon, France. [7] Laboratory for Molecular and Cellular Technology, Queen Astrid Military Hospital, 1120 Brussels, Belgium. [8] Service de Neurochirurgie, Chirurgie du Rachis et de la Moëlle Épinière, Hôpital Pierre Wertheimer, Hospices Civils de Lyon, Lyon, France. [9] Centre of Research and Innovation in Clinical Pharmaceutical Sciences, Lausanne University Hospital, Lausanne, Switzerland. [10]These authors contributed equally: J.-P. Pirnay, G. Resch. ✉email: tristan.ferry@univ-lyon1.fr

Bone and joint infections (BJI) are one of the most difficult-to-treat bacterial infection, especially in the era of antimicrobial resistance. Lytic bacteriophages (phages for short) can rapidly and selectively target and kill bacteria whilst producing new phage particles in an exponential and self-sustained reaction. Accordingly, lytic phages are considered to have a high therapeutic potential for the treatment of bacterial infectious diseases[1]. Moreover, regarding their often-described synergistic anti-biofilm activity when combined with antibiotics, phages are considered as particularly promising adjuvant therapy for the treatment of complex BJI[2,3]. Currently, phage Active Pharmaceutical Ingredients (pAPIs) production follows requirements of quality and safety, which guarantee adequate composition and acceptable levels of residual contaminants[1,4]. Here we report on the management of a patient with a pandrug-resistant *Pseudomonas aeruginosa* spinal abscess who was treated with surgery and personalized phage therapy that was possible thanks to a unique academic collaboration.

## Results

A 74-year-old man with melanoma treated with anti-PD1 (pembrolizumab) experienced a catheter-related bacteremia due to multidrug-resistant *P. aeruginosa* in December 2017 (Fig. 1). He was treated successfully with colistin and meropenem that he received during 11 days. He developed spinal pain during summer 2018 and spondylodiscitis with spinal abscess was diagnosed (Figs. 1 and 2A, B) in December 2018. Aspiration revealed pan drug-resistant *P. aeruginosa* in culture, with resistance to all antibiotics including ceftazidime/avibactam (MIC 64 mg/L), ceftolozane/tazobactam (MIC > 256 mg/L) and colistin (MIC 8 mg/L). He received in a French hospital colistin (colistimethate sodium; 3 MUI/8 h) and rifampin 900 mg once a day, and as he developed acute mild kidney injury, antibiotics were rapidly stopped. The patient was admitted in our referral center for the management of

complex BJI (http://www.crioac-lyon.fr) with severe pain. He was bedridden and required continuous infusion of opioid. As no active antibiotic was available for the treatment, phage therapy, that was already used in our institution, was envisaged as salvage therapy. The strain was sent to two different private companies (in France and in the USA) to test their phages in development, and to a European military laboratory that have purified phages. Unfortunately, the strain was fully resistant to all available phages. Therefore, we developed a unique academic collaboration between universities and hospitals located in three different European countries (Switzerland, Belgium, and France) that allowed us to identify three different lytic phages active on the patient initial *P. aeruginosa* isolate, to produce them separately, and to administer them as a pre-assembled personalized three-phage cocktail to the patient, in the shortest possible time (see methods, Fig. 3A–F and Tables 1 and 2).

In collaboration, we proposed phage therapy combined with a surgical staged strategy (approved by the ethic committee of our hospital and by the French national health care authority; performed after the patient gave informed consent). Indeed, a step-by-step approach, or staged surgical strategy, is usually performed in this type of infection, with two steps pursuing different aims. In complex infections requiring implementation of definitive implants, a first stage consisting of debridement, abscess evacuation and antibiotics is required to treat the infection. Then a second stage is needed, usually after at least 2 weeks of antibiotic window, to permanently stabilize the spine with definite osteosynthesis. During the second stage, new bacterial samples are usually harvested to check for microbiological success, and to rule out superinfection. More specifically, in the case presented here, we performed a first stage consisting of open surgery, L2-L3 and L3-L4 disc debridements (discectomy), local administration of the three-phage cocktail, and a partial stabilization, bridging the infected area with osteosynthesis from T11-T12-L1 to L5-sacrum

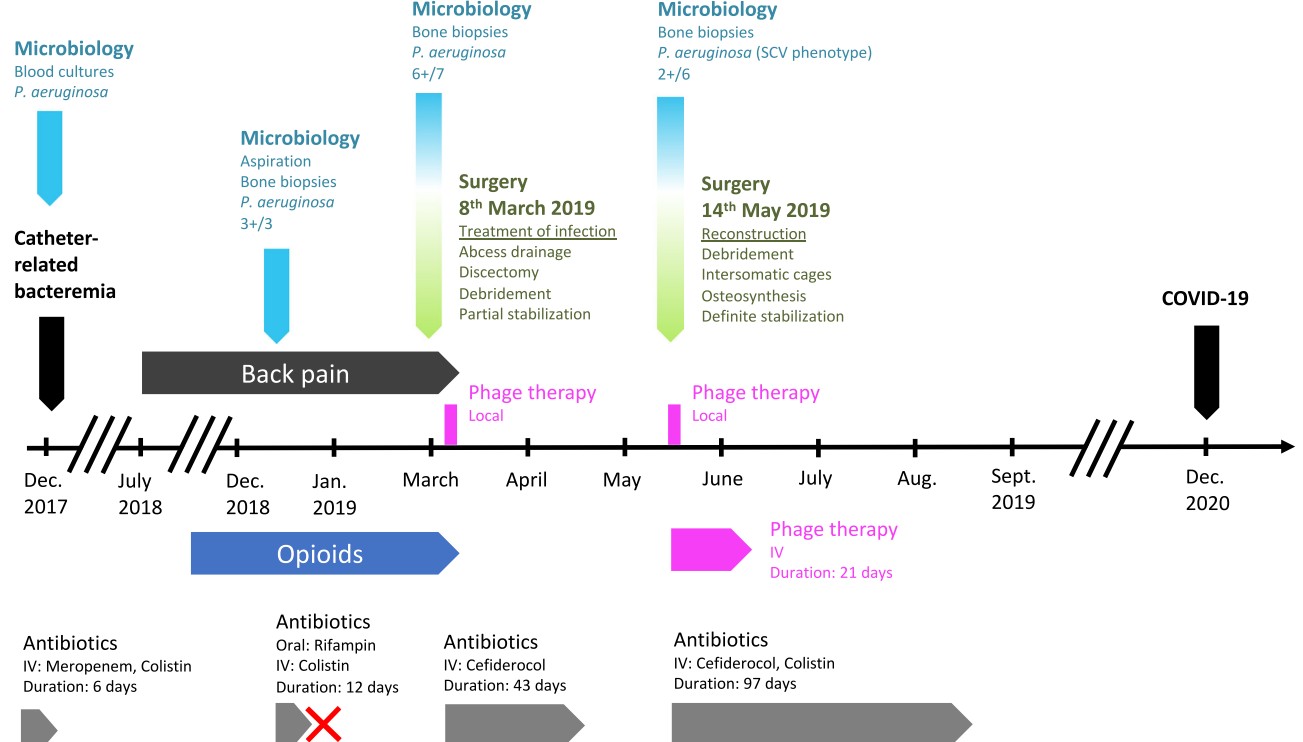

**Fig. 1** Timeline of the most relevant surgical procedures (green), microbiological results (light blue), opioid therapy (dark blue), antibiotic therapies (dark gray), and phage therapy (magenta). Clinical events are indicated in black. The red cross indicated withdrawal of antibiotics due to the occurrence of a serious adverse event (mild kidney injury).

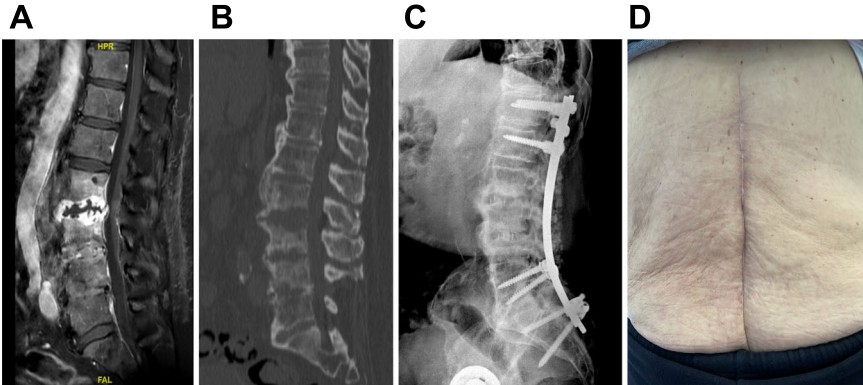

**Fig. 2 Imaging of the patient before and after treatment. A** Gadolinium T1-weighed MRI of the spine showing L2-L3 disc abscess and L3-L4 spondylodiscitis; **B** CT-scan showing mirror-like bone destruction from either side of the L2-L3 abscess; **C** X-ray performed at the end of the follow-up showing no loosening of the spinal osteosynthesis and the adequate position of the intersomatic cages at L2-L3 and L3-L4 level with anterior bone fusion; **D** Local aspect of the lumbar scar at the end of the follow-up, showing no inflammation nor discharge.

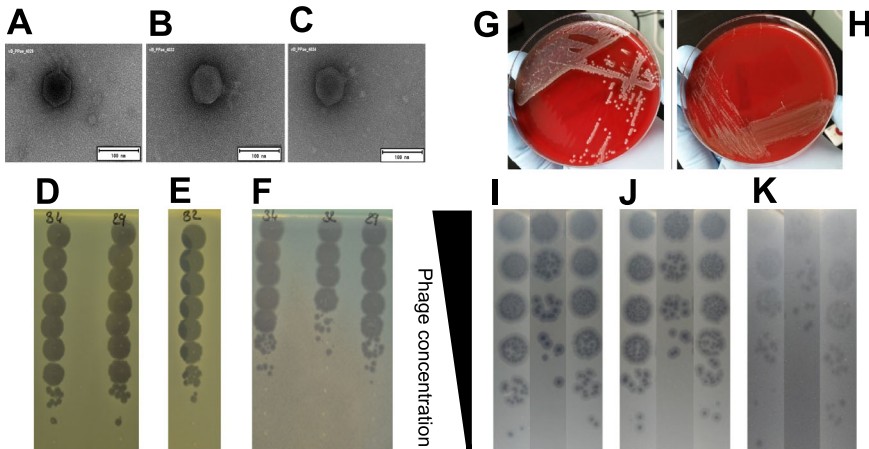

**Fig. 3 Morphologic description of the phages used to treat the patient and description of their activity on the patient's strains. A** EM micrographs of phage vB_PaeP_4029; **B** EM micrographs of phage vB_PaeP_4032; **C** EM micrographs of phage vB_PaeP_4034; **D** Phagogram (spot test) of phages vB_PaeP_4034 and vB_PaeP_4029 on production strain PAO1; **E** Phagogram (spot test) of phage vB_PaeP_4032 on production strain ATCC-15442-MINI; **F** Phagogram (spot test) of phages vB_PaeP_4034, vB_PaeP_4032, and vB_PaeP_4029 on the patient strain; **G** One of the two *P. aeruginosa* morphotypes cultured from a bone sample taken from the second-stage surgery procedure, that was close to the initial strain; **H** One of the two *P. aeruginosa* corresponds to a stable Small Colony Variant (SCV) morphotype; **I** Visualization of Plaque Forming Units (PFU) of the phage vB_PaeP_4029 on the patient's strains isolated before phage therapy or during the second-stage surgery procedure; **J** Visualization of Plaque Forming Units (PFU) of the phage vB_PaeP_4032 on the patient's strains isolated before phage therapy or during the second-stage surgery procedure; **K** Visualization of Plaque Forming Units (PFU) of the phage vB_PaeP_4034 on the patient's strains isolated before phage therapy or during the second-stage surgery procedure.

(to avoid neurological complications), and systemic cefiderocol antibiotherapy for 6 weeks. This was followed by a second stage consisting of definite reconstruction with removal of the partial stabilization osteosynthesis, inter body fusion cages in L2-L3 and L3-L4, and spinal osteosynthesis outside the septic location, from T12-L1 to L5-S1, to finalize consolidation. For the first stage, magistral preparation of the phage cocktail was done in real time by our hospital pharmacist. During the first stage, the phage cocktail (dilution in 7 mL; final phage titer of $10^6$ pfu/mL) was administered locally at L2-L3 and L3-L4 after abscess evacuation, debridement, and lavage with bicarbonate solution. Additive quality controls (QCs) were performed by the French hospital pharmacy that included sterility test for bacterial and fungal contamination and endotoxin concentration. Of note, six out of the seven microbiological samples performed during surgery were positive to *P. aeruginosa* (same antibiogram). Cefiderocol was started intravenously after the surgery, 2 g during 3 h every 8 h (6 g/day) for a duration of 6 weeks. Quickly, the pain was significantly reduced. During the treatment, the patient experienced

*Clostridioides difficile* diarrhea, but no other serious adverse event was reported. Two weeks after the end of the first stage (two months after the initial surgical debridement), and then 2 weeks after the withdrawal of cefiderocol, the second stage was performed. The patient had neither systemic (no fever, CRP 10 mg/L) nor clinical signs of persistent infection. The same phage cocktail was locally administered (same volume and phage concentration) before insertion of the intersomatic cages at L2-L3 and L3-L4 level. Cefiderocol was started again intravenously pending the culture results. However, *P. aeruginosa* still grew in culture from bone biopsy (only two out of the six samples that were collected) with a small colony variant phenotype, but remained susceptible to the phage cocktail and cefiderocol (Fig. 3G–K, Table 2). Whole genome sequencing of this strain identified numerous acquired genes accounting for this high level of resistance to antibiotics (Table 3). Although the strain had become resistant to this antibiotic, colistin (colistimethate sodium) was added intravenously at a dose of 2 MUI/8 h (6 MUI/day) to potentially synergize with cefiderocol[5]. As the cultures

revealed persistence of the *P. aeruginosa*, phages were also added intravenously over 3-h infusions (30 mL, phage titers $10^6$ pfu/mL) every day for 21 days (Fig. 1). Under this treatment, the patient experienced abdominal pain related to gall stone migration and relapsing *C. difficile* infection. No adverse event potentially related to phage therapy was noticed. Antibiotics (cefiderocol and colistin) were stopped at 3 months. The outcome was favorable during the follow-up (21 months), without implant loosening nor clinical signs of infection (Fig. 2C, D), and the patient was walking without pain (Supplementary Movie 1). Unfortunately, he died from severe COVID-19 pneumonia responsible for refractory hypoxia in December 2020.

## Discussion

Phage therapy is an emerging option for complex BJI, especially in the era of worldwide dissemination of resistance[1]. At the present time, no phages are commercially available, and some companies recently performed clinical trials in the setting of burn patients or in patients with bacteremia, with no safety signal[6,7]. Few data have been published about phage therapy in patients with BJI, especially in patients with spinal infection due to multidrug-resistant *P. aeruginosa*. Some patients were treated using phages under development by private companies, and others were treated by academic structures, due to the lack of availability or lack of activity of phages in development[8–13].

Despite not being able to strictly determine the relative contribution of phages and antibiotics in the improvement of the patient in the present case study, we believe that personalized phage therapy is a potential adjuvant treatment of complex BJI, in particular due to pandrug-resistant *P. aeruginosa*. Indeed, without the use of antibiotics that have anti-biofilm activity (i.e., rifampin in staphylococcal implant-associated infections, or a fluoroquinolone in Gram-negative infections), the cumulative probability of failure is considered as very high in patients with implant-associated infections. In the case presented here, a fluoroquinolone was not usable due to the multidrug resistance profile of the strain, and the potential anti-biofilm activity of the phages that have been used to treat the patient probably helps for the cure[14,15].

In the present case, a limiting factor was the time to screen, produce, and purify the bacteriophages (in total 3 months) before a pre-assembled, personalized, and targeted phage cocktail could be administered to the patient as an adjuvant to surgery and antibiotics. Consequently, having access to a collection of purified phages, which could be used in a short delay, could be extremely important for future treatments of such severe bacterial infections. At the very least, the present case study has once again demonstrated the safety of purified phages, administered multiple times locally and intravenously. Accordingly, regarding the highly personalized approach needed for phage therapy, academic collaborations, in addition to industrial players, are very valuable to develop the field. Indeed, as demonstrated here, a unique European academic collaboration in the context of a life-saving treatment provided again important data that hopefully will facilitate the establishment and conduct of clinical trials in a foremost relevant indication.

## Methods

**Ethical approval**. The patient has consented to our use of samples and data in research and to the publication of clinical data, the photo and the video presented here. He also gave written informed consent to receive phage therapy in accordance with CARE guidelines and the principles of the Declaration of Helsinki. The ethics committee of "Hospices Civils de Lyon" concluded that phage therapy was ethically justified in this clinical situation (19-162).

**Table 1 Phagogram (Efficiency of Plating [EOP]) of the non-purified phages on the patient strain.**

|                              | vB_PaeP_4029      | vB_PaeP_4032      | vB_PaeP_4034        |
| ---------------------------- | ----------------- | ----------------- | ------------------- |
| Titer on production strain   | $8 \times 10^9$   | $2 \times 10^9$   | $8 \times 10^9$     |
| Titer on patient strain      | $4 \times 10^8$   | $2 \times 10^7$   | $6 \times 10^8$     |
| EOP score on patient strain  | $5 \times 10^{-2}$ | $1 \times 10^{-2}$ | $7,5 \times 10^{-2}$ |

**Table 2 Determination of EOP scores of the three phages on the patient's strains isolated before phage therapy or during the second-stage surgery procedure (2 morphotypes, including one SCV morphotype).**

| Strain    | Initial strain |  |  | Second stage surgery Morphotype 1 |  |  | Second stage surgery Morphotype 2 |  |  |
| --------- | ----------------- | ----------------- | ----------------- | ----------------- | ----------------- | ----------------- | ----------------- | ----------------- | ----------------- |
| Phage     | vB_PaeP_4029      | vB_PaeP_4032      | vB_PaeP_4034      | vB_PaeP_4029      | vB_PaeP_4032      | vB_PaeP_4034      | vB_PaeP_4029      | vB_PaeP_4032      | vB_PaeP_4034      |
| EOP score | $5.6 \times 10^{-2}$ | $1.2 \times 10^{-3}$ | $1.4 \times 10^{-1}$ | $1 \times 10^{-1}$ | $1.1 \times 10^{-3}$ | $1.3 \times 10^{-1}$ | $1.5 \times 10^{-2}$ | $8.3 \times 10^{-5}$ | $1.1 \times 10^{-2}$ |

**Table 3 List of chromosomal mutations identified in the genome of the SCV morphotype (n°2) when compared to the genome of morphotype n°1.**

| Gene                | Protein                                         | Mutation    | Effect on the protein                    | Impact on the protein function                            |
| ------------------- | ----------------------------------------------- | ----------- | ---------------------------------------- | --------------------------------------------------------- |
| Hypothetical protein | Unknown                                        | 99T > C     | synonymous variant                       | Unlikely                                                  |
| *sdh*A              | Succinate dehydrogenase flavoprotein subunit    | 1673A > G   | missense variant: His558Arg              | Unlikely: same family of aminoacids                       |
| *ser*A              | Phosphoglycerate deshydrogenase                 | 40T > C     | missense variant: Phe14Leu               | Possible: different families of aminoacids                |
| Hypothetical protein | Unknown                                        | 739A > C    | missense variant: Thr247Pro              | Unknown                                                   |
| *pbu*E              | Purine efflux pump PbuE                          | 605G > AT   | missence variant: insertion of Arg in 202 | Unlikely (end of protein)                                 |
| *nod*D2             | Nodulation protein                              | 303G > T    | synonymous variant                       | Unlikely                                                  |
| *lgr*D              | non-ribosomal peptide synthetase                | 2668A > T   | stop gained Lys890*                      | Possible: truncated protein of 890 amino acids versus 991 |

Variant calling was performed using the Snippy software (https://github.com/tseemann/snippy) after annotation of the reference genome of morphotype n°1 (Prokka software).

**Phagogram**. Phagograms were performed as follow: 15 mL of pre-warmed (45–50 °C) LB containing 0.75% (w/v) of agar (LB soft-agar) were seeded with 200 µL overnight culture of each bacterial strain tested and poured into 90 mm round Petri dishes. After solidification of the LB soft-agar at room temperature, 5 µL drops of 10-time serially diluted stock suspensions of each phage was deposited on top of the LB soft-agar layer, the most concentrated suspension at the top and the most diluted one at the bottom of the Petri dish. After drying the drops at room temperature, the Petri dishes were incubated aerobically at 37 °C over-night. The next day, the Petri dishes were checked by eye for lysis zones.

**Phage characterization**. Phagogram was performed with >100 *P. aeruginosa* phages from the Lausanne University phage collection to find active phages on the patient's strain. The genomic DNA of three active phages (named vB_PaeP_4029, vB_PaeP_4032 and vB_PaeP_4034) were fully sequenced. Phage genomic libraries were prepared with an optimized protocol and standard Illumina adapter sequences, and sequencing was performed with Illumina technology, NovaSeq 6000 (read mode 2 × 150 bp) at Eurofins Genomics Germany GmbH (Ebersberg, Germany). Reads were assembled into contigs using the PATRIC v3.6.12 pipeline for assembly with parameters "trim read before assembly = TRUE", "min contig length = 1000", and "min contig coverage = 100" and contigs were annotated using the PATRIC v3.6.12 pipeline for annotation following the "bacteriophage recipe" with default parameters (https://www.patricbrc.org/). The genomes of the three phages were 72,063 bp in length. Using blastn (https://blast.ncbi.nlm.nih.gov/), the closest neighbor of vB_Pae4029 was phage vB_PaeP_PYO2 (Genbank accession number MF49236.1) with 100% coverage and 99.72% identities. The closest neighbor of vB_PaeP_4032 and vB_PaeP_4034 was phage PEV2 (Genbank accession number KU948710.1) with 100% coverage and 99.94% identities and 100% coverage and 99.95% identities, respectively. Phage vB_PaeP_4029 had 99.86% identities over 99% coverage with phage vB_PaeP_4032 and 99.73% identities over 100% coverage with phage vB_PaeP_4034. Phage vB_PaeP_4032 and vB_PaeP_4034 differed by five single nucleotide polymorphisms. The phages, which looked as podoviruses on the Electron Microscopy (EM) micrographs (Fig. 3A–C) were further classified in the family *Schitoviridae*, genus *Litunavirus* based on their genome sequences. The lytic nature of the three phages has been verified using the repository of PhageAI (https://app.phage.ai/phages/), which confirmed their suitability for use in a phage therapy protocol. Moreover, the total absence on the three genomes of genes related to 31,552 genes encoding for known and predicted virulence factors listed in the Virulence Factors Database (VFDB, http://www.mgc.ac.cn/VFs/main.htm) confirmed their good safety profile. The phagogram and the Efficiency of Plating tests performed as previously described[16] on the patient and production strain for the three unpurified phages revealed lytic activity (Fig. 3D–F and Table 1).

**Phage production**. Production of the pAPIs, in compliance with a Belgian pharma-copeial monograph describing the production process and QC system for incorporation in magistral preparations, was done in the laboratory of the Queen Astrid military hospital in Brussels in collaboration with pharmacists from the Croix-Rousse hospital (Hospices Civils de Lyon)[4,16], and under the supervision of the French national health care authority (Agence Nationale de Sécurité du Médicament et des produits de santé; ANSM). Of note, the monograph received on 10 January 2018 a formal positive advice from the Belgian Minister of Public Health asked the Federal Agency for Medicines and Health Products (FAMHP). It was conceived by representatives of the Queen Astrid Military Hospital located (QAMH) in Brussels, the FAMHP and Sciensano, formerly known as the Belgian Scientific Institute of Public Health. Here the three *P. aeruginosa* phages were produced using *P. aeruginosa* host strains PAO1 (for phages vB_PaeP_4029 and vB_PaeP_4034) and ATCC® 15442™ (for phage vB_PaeP_4032), according to the monograph for phage APIs. Phages were propagated using the double-agar overlay method in animal product-free growth media with alternative protein source (APS-LB broth, BD) following working instructions (WI) developed by QAMH. Phages were mixed with corresponding bacterial host strains at the expected multiplicity of infection in the range of $10^{-3}$–$5 \times 10^{-4}$ and APS-LB 0.6% agar at 45 °C in the final volume of 12 mL and plated on 12 mm square plates filled with 55 mL APS-LB 1.5% agar. Plates were incubated overnight at 37 °C. The upper layer was scraped from the plates and centrifuged at $6000 \times g$ for 20 min. Obtained supernatants were filtered through 0.45 µm and 0.22 µm polyether sulfone (PES) filters. Further, phages were pelleted by centrifugation at $35,000 \times g$ for 60 min. The resulting phage pellet was resuspended in 10 times less volume of Dulbecco's Phosphate Buffered Saline (DPBS) buffer to obtain purified phage stocks at high titers, i.e., in the range of $10^{10-11}$ pfu/mL. Next, phage production process was continued in the clean room facility where phage stocks were conditioned as APIs by diluting to a final concentration of $10^{9-10}$ pfu/mL in DPBS, followed by filtration through 0.22 µm PES filters, endotoxin purification using EndoTrap® columns (Lionex, Germany) and final filtration through 0.22 µm medical grade polyvinylidene fluoride filters. Samples of each of the three phage APIs were sent to Sciensano for QC testing, including determination of pH, endotoxin level (EU/mL), and microbial burden. The pH values for the phage APIs ranged from 7.29 to 7.35, the endotoxin content endotoxin concentration ranged from 134 to 3400 UI/mL and no bacterial growth was observed in any of the samples. Based on the above results the three phage APIs were approved by Sciensano and deemed safe for various application in humans taking into account limits of endotoxin level for each specific administration route[4,16].

**Temporary authorization for use of cefiderocol**. We obtained from the ANSM a temporary authorization for use (Autorisation Temporaire d'Utilisation, ATU) of cefiderocol, a new cephalosporin evaluated in clinical trials[17], which showed activity against the patient's *P. aeruginosa* strain (MIC = 1 mg/L, broth microdilution).

**Reporting summary**. Further information on research design is available in the Nature Research Reporting Summary linked to this article.

## Data availability
The full genome sequence and associated data for phage vB_PaeP_4029, vB_PaeP_4032, and vB_PaeP_4034 were deposited in GenBank (https://www.ncbi.nlm.nih.gov/genbank/) under BioProject number PRJNA691459. The genome accession numbers are ON815901, ON815902, and ON815903, respectively. The BioSample numbers are SAMN28810961, SAMN28810962, and SAMN28810963, respectively. The raw reads were deposited under accession numbers SRX15796894, SRX15796895, and SRX15796896, respectively.

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

## Acknowledgements
We acknowledge Cyrille Confavreux (who participated in the patient care before admission to our hospital) and Aurélie Marchet (who participated in the phage

characterization). We also acknowledge the research directory of Hospices Civils de Lyon and the Hospices Civils de Lyon foundation, which support the development of phage therapy in the Lyon university hospitals through the PHAGE*in*LYON program.

## Author contributions

T.F. designed the therapeutic approaches, managed the patient and organize the treatment among all participants, and interacted directly with the French health authority; C.K., F.L., and C.A.G. performed microbiological analyses for the diagnosis of infection; G.L. participated in the performance of quality controls and performed phage preparations; M.M., S.D., and J.P.P. participated to the purification of phages; T.P. and C.B. participated to the patient care; G.R. found active phages and participated to their characterization.

## Funding

GR lab supported the cost of phages characterization; QAMH supported the cost of phage purification and QC. M.M. is supported by the Royal Higher Institute for Defense (Belgium).

## Competing interests

The authors declare no competing interests.
