## [Peer Review File · Nature Communications]

REVIEWER COMMENTS

Reviewer #1 (Remarks to the Author):

This is a well-done and interesting case report of a challenging case of a patient with pan-resistant *Pseudomonas* spinal infection. There are several noteworthy results which contributes to the field and demonstrates the complexities and nuances of the role of phage therapy in very challenging cases.

Several notable aspects of this case:

- 1- Combination of direct administration and systemic phages
- 2- Done in conjunction with aggressive staged surgical approach
- 3- Done in conjunction with standard of care antibiotics
- 4- Highly personalized with evolving plan tailored to the clinical case
- 5- Demonstrates collaborative effort to newly source a phage

One issue for this case is lack of proof that the phages provided added benefit to the conventional practice of staged surgical approach in conjunction with antibiotics, which the authors acknowledge. This same issue applies to any treatment which is adjunctive. However, I wonder if that authors can provide any perspective on how a patient with spinal hardware implanted in the presence of positive *P. aeruginosa* cultures can be expected to perform? Can expected success or failure rates be estimated based on historical data to act as a control?

Other clarifications the authors can provide:

- 1- Was there quantification of the bacterial growth at the time of the 2 surgeries? (1+, 2+, etc?).
- 2- How was the timing of the second stage determined? Was it based on clinical signs, symptoms and imaging or decided a priori?
- 3- Did the authors use any particular indicators to help determine the duration of phage and antibiotic treatments?

4- Were there any “before” clinical photos that can be provided for comparison to Panel O?

Minor comments:

Line 15: “2-stage approach” typically refers to specifically to a “2-stage exchange” in the musculoskeletal infection literature, wherein infected hardware is removed and exchanged for new hardware in a staged fashion. I would use the term “staged approach” instead.

Line 17: Add “was” cured

Line 34-38: Was the initial *P. aeruginosa* isolate susceptible to meropenem and colistin? If not, was there an attempt to document successful clearance of the bacteremia?

Line 41: Use “aspiration” instead of puncture

Line 57: Use “In collaboration, we” instead of “We collegially”

Line 60 and throughout manuscript: Use “disc” instead of “disca”

Line 66: Remove “extemporaneously” and use “in real time”

Line 72: “over” 3 hours

Line 77: Use “concentration” instead of “dilution”

Line 98-99: use “no phages are commercially available” instead of “on the market”

Line 100-101: Sentence with “Few data....” Is this true? There are many case reports on phages in BJI.

Line 204: What is meant by “mirror-like” here?

Line 224: Correct the spelling of lumbar

Reviewer #2 (Remarks to the Author):

1.The authors claim (line 100) that: "few data have been published about phage therapy in patients with BJI". However, four reviews on that subject appeared just in the recent months: 1. Genevieve J et al EFORT Open Rev 2021,10,1148; 2.Ferry T et al Viruses 2021,13,2414; 3.van Nieuwenhuyse B et al Viruses 2021,13,1898; 4.Walter N et al Orthopede 2022,51138. Does this report provide any important new information that would justify its presentation to to a scientific and medical community? This reviewer finds it difficult to agree on that.

2.line 45: "the patient developed acute kidney injury and the antibiotics were rapidly stopped". This condition (also referred to as acute renal failure) is a medical emergency usually requiring dialysis. However, there is no comment on how this grave complication was treated. If indeed hemodialysis was applied then some discussion on its possible influence on phage therapy should be added.

3.line 99: "some companies recently performed clinical trials".

Yes, and the results do not provide evidence for the effectiveness of phage therapy - shouldn't that fact be mentioned?

4.line 95: "the patient died of COVID" - if the authors believe that their report provides support for clinical application of phage therapy then some discussion to exclude phage contribution to the patient's death should be added.

5.line 112: "European academic collaboration is crucial to develop the field". Of course, but a clear statement that such collaboration should primarily involve clinical trials is lacking.

“Response to referees” letter above the manuscript NCOMMS-22-01270-T

REVIEWER COMMENTS

Reviewer #1 (Remarks to the Author):

This is a well-done and interesting case report of a challenging case of a patient with pan-resistant *Pseudomonas* spinal infection. There are several noteworthy results which contributes to the field and demonstrates the complexities and nuances of the role of phage therapy in very challenging cases.

Several notable aspects of this case:

- 1- Combination of direct administration and systemic phages
- 2- Done in conjunction with aggressive staged surgical approach
- 3- Done in conjunction with standard of care antibiotics
- 4- Highly personalized with evolving plan tailored to the clinical case
- 5- Demonstrates collaborative effort to newly source a phage

Answer: Thanks for these comments.

One issue for this case is lack of proof that the phages provided added benefit to the conventional practice of staged surgical approach in conjunction with antibiotics, which the authors acknowledge. This same issue applies to any treatment which is adjunctive. However, I wonder if that authors can provide any perspective on how a patient with spinal hardware implanted in the presence of positive *P. aeruginosa* cultures can be expected to perform? Can expected success or failure rates be estimated based on historical data to act as a control?

Answer: Thanks for this comment. This is an important question. There is no data about the risk of failure in patients with *P. aeruginosa* persistent spinal infection at the time of reconstruction, especially as *P. aeruginosa* is not a common pathogen in this setting, and as patients who need reconstruction are only part of the patients who experienced spondylodiscitis. However, there are some data that indicating that the risk of failure is probably high in our patient. First, positive culture at the time of reconstruction has to be considered as an implant-associated infection, for which the use of drugs that have anti-biofilm activity is crucial to improve the prognosis. Fluoroquinolone is a family of antibiotics that is considered to have anti-biofilm activity against Gram-negative bacteria. In the paper published by Köder et al. in 2020 in the journal “Infection”, the authors reported that patients with spinal implant-associated infections treated without biofilm-active antibiotics was associated with a worse outcome, and with a higher post-operative pain intensity in comparison with patients treated with biofilm-active antibiotics. Of note, few patients were infected with Gram-negative bacteria in this study, none of them were infected with *P. aeruginosa*, and all Gram-negative bacteria were associated with polymicrobial infection. Despite the limitation of the interpretation of the results in this study dedicated to spinal implant-infection, our group published concordant results in a retrospective cohort study that has been performed in patients with implant-associated *P.*

aeruginosa infection (Cerioli et al. *Frontiers Medicine* 2020). Among the 90 patients included in the study, 15 had spinal implant associated infection due to *P. aeruginosa*. This study indicated that the use of fluoroquinolone influenced the prognosis, with a cumulative probability of failure of 90% for patients for whom a fluoroquinolone was not prescribed (mainly due to antimicrobial resistance) (Figure 1C). In the case presented here, no fluoroquinolone was used, as the strain responsible for the infection was a multidrug resistant strain. We added in the discussion section of the manuscript the following sentence: "Indeed, without the use of antibiotics that have anti-biofilm activity (i.e. rifampin in staphylococcal implant-associated infections, or a fluoroquinolone in Gram-negative infections), the cumulative probability of failure is considered as very high in patients with implant-associated infections. In the case presented here, a fluoroquinolone was not usable due to the multidrug resistance profile of the strain, and the anti-biofilm activity of the phages that have been used to treat the patient potentially helps for the cure." We added the references Köder et al. and Cerioli et al.

Other clarifications the authors can provide:

1- Was there quantification of the bacterial growth at the time of the 2 surgeries? (1+, 2+, etc?).

Answer: Bacterial quantification was not mentioned in the manuscript because analytical procedures for bone samples are not standardized to provide reliable quantification of bacteria (important variations of the sample size which is diluted in sterile water for grinding, homogenization is often difficult etc.). However, the number of positive samples out of the total number of samples taken during the surgery is more interesting. Indeed, 6/7 samples taken during the first surgery were positive versus 2/6 samples taken during the second surgery. We added these informations in the manuscript.

2- How was the timing of the second stage determined? Was it based on clinical signs, symptoms and imaging or decided a priori?

Answer: Thanks for this question. The timing of the second stage is mainly based on our experience, so decided a priori. It is usual to perform the second stage after an antibiotic window of at least two weeks. Of course, in case of clinical signs of relapse during the initial antimicrobial therapy or during the antibiotic window, a new debridement has to be done, and the reconstruction has to be delayed.

3- Did the authors use any particular indicators to help determine the duration of phage and antibiotic treatments?

Answer: Thanks for this comment. We are using the data from the literature and different guidelines to determine the duration of antimicrobial therapy. In patients with a native hematogenous spinal infection, a duration of 6 weeks is usual. In patients with an implant-associated infection, the usual duration is frequently 3 months. Some patients with persistent infection at the time of reconstruction could be qualified for indefinite suppressive antimicrobial treatment (i.e. a life-long daily intake of antibiotics), that was not possible in the case presented here, as no oral options were available.

4- Were there any “before” clinical photos that can be provided for comparison to Panel O?

Answer: Thanks for this comment. In fact, a photo of the back of the patient before the surgery was not provided as clinical symptoms during hematogenous spinal infection are most of time the pain, without any detectable abnormalities at the inspection of the skin, as there is no disruption of the posterior fascia that contain the infection. After a surgery, as a disruption of the fascia has been done by the surgeon to access to the spine, it is of importance to check the scars for inflammation or occurrence of a fistula, as it could be a sign of relapse.

Minor comments:

Line 15: “2-stage approach” typically refers to specifically to a “2-stage exchange” in the musculoskeletal infection literature, wherein infected hardware is removed and exchanged for new hardware in a staged fashion. I would use the term “staged approach” instead.

Answer: Thanks for this comment. The modification has been done.

Line 17: Add “was” cured

Answer: Thanks for this comment. The modification has been done.

Line 34-38: Was the initial *P. aeruginosa* isolate susceptible to meropenem and colistin? If not, was there an attempt to document successful clearance of the bacteremia?

Answer: Thanks for this comment. The isolate was resistant to meropenem. The probability to obtain the cure at this stage was considered to be low.

Line 41: Use “aspiration” instead of puncture

Answer: Thanks for this comment. The modification has been done.

Line 57: Use “In collaboration, we” instead of “We collegially”

Answer: Thanks for this comment. The modification has been done.

Line 60 and throughout manuscript: Use “disc” instead of “discal”

Answer: Thanks for this comment. The modification has been done.

Line 66: Remove “extemporaneously” and use “in real time”

Answer: Thanks for this comment. The modification has been done.

Line 72: “over” 3 hours

Answer: Thanks for this comment. The modification has been done.

Line 77: Use “concentration” instead of “dilution”

Answer: Thanks for this comment. The modification has been done.

Line 98-99: use “no phages are commercially available” instead of “on the market”

Answer: Thanks for this comment. The modification has been done.

Line 100-101: Sentence with “Few data....” Is this true? There are many case reports on phages in BJI.

Answer: Thanks for this comment. Yes indeed, there are many case reports in patients with BJI, but in very diverse clinical situation. We added “especially in patients with spinal infection due to multidrug-resistant *P. aeruginosa*.”

Line 204: What is meant by “mirror-like” here?

Answer: Thanks for this comment. It is the typical description on CT scan of spondylodiscitis. There is a bone destruction from either side of the infected disc. We added “from either side” in the manuscript.

Line 224: Correct the spelling of lumbar

Answer: Thanks for this comment. The modification has been done.

Reviewer #2 (Remarks to the Author):

1.The authors claim (line 100) that: "few data have been published about phage therapy in patients with BJI". However, four reviews on that subject appeared just in the recent months: 1. Genevieve J et al EFORT Open Rev 2021,10,1148; 2.Ferry T et al Viruses 2021,13,2414; 3.van Nieuwenhuysse B et al Viruses 2021,13,1898; 4.Walter N et al Orthopede 2022,51138. Does this report provide any important new information that would justify its presentation to to a scientific and medical community? This reviewer finds it difficult to agree on that.

Answer: Thanks for this comment. Yes indeed, there are many case reports in patients with BJI, but in very diverse clinical situation, and not in the setting presenting here. We added “especially in patients with spinal infection due to multidrug-resistant *P. aeruginosa*.” In the manuscript. We also added the following references proposed by the reviewer: Genevière et al. and Ferry T et al.. We did not include the reference van Nieuwenhuysse et al. as it is a case report with phages targeting *S. aureus*. We also added another new reference: Pirnay et al. FEMS 2022 as it is an important review in the field of phage therapy, that is not restrictive to the field of BJI.

2.line 45: "the patient developed acute kidney injury and the antibiotics were rapidly stopped". This condition (also referred to as acute renal failure) is a medical emergency usually requiring dialysis. However, there is no comment on how this grave complication was treated. If indeed hemodialysis was applied then some discussion on its possible influence on phage therapy should be added.

Answer: Thanks for this comment. It was a mild reversible kidney injury (no dialysis was required), before the phage administration, associated with the prescription of colistin.

3.line 99: "some companies recently performed clinical trials".
Yes, and the results do not provide evidence for the effectiveness of phage therapy - shouldn't that fact be mentioned?

Answer: Thanks for this comments. In the cited clinical trials (Phase I/II), the main objective was the safety, and the feasibility of phage therapy. We agree that another clinical trial failed to demonstrate the efficacy of phage therapy, but it was in another setting: diarrhea in children from Bangladesh. The cited articles brought important data about the safety of phage therapy, as they did not record any safety signal. We added in the manuscript: "with no safety signal".

4.line 95: "the patient died of COVID" - if the authors believe that their report provides support for clinical application of phage therapy then some discussion to exclude phage contribution to the patient's death should be added.

Answer: Thanks for this comment. We gave more information about the COVID infection: the patient developed typical COVID pneumonia with severe hypoxia, that was not related to the application of phages that was done more than 18 months before. We modified the manuscript as following: "... died from severe COVID-19 pneumonia responsible for refractory hypoxia".

5.line 112: "European academic collaboration is crucial to develop the field". Of course, but a clear statement that such collaboration should primarily involve clinical trials is lacking.

Answer: Thanks for this comment. We added the following sentence: "Such collaboration should primarily involve for compassionate salvage treatment in severe bacterial infections and has to facilitate the built and the performance clinical trials in the most relevant indications."